# Antioxidant Effect of a Dietary Supplement Containing Fermentative S-Acetyl-Glutathione and Silybin in Dogs with Liver Disease

**DOI:** 10.3390/vetsci10020131

**Published:** 2023-02-08

**Authors:** Elisa Martello, Francesca Perondi, Donal Bisanzio, Ilaria Lippi, Giorgia Meineri, Valeria Gabriele

**Affiliations:** 1Division of Epidemiology and Public Health, School of Medicine, University of Nottingham, Nottingham NG 1PB, UK; 2Department of Veterinary Science, University of Pisa, San Piero a Grado, 56122 Pisa, Italy; 3RTI International, Washington, DC 20005, USA; 4Department of Veterinary Sciences, University of Turin, Largo Paolo Braccini, 10095 Grugliasco, Italy; 5Candioli Pharma S.R.L., S.da Antica di None, 10092 Beinasco, Italy

**Keywords:** oxidative stress, dog, liver disease, oral administration, GSH

## Abstract

**Simple Summary:**

The use of a dietary supplement containing S-acetyl-glutathione (SAG), silybin, and other antioxidant ingredients increased the level of erythrocyte glutathione (GSH) and improved key biochemical parameters in dogs with liver disease.

**Abstract:**

Oxidative stress is often involved in liver disease progression. Liver is the primary site for the synthesis of glutathione (GSH), the major intracellular antioxidant. GSH erythrocyte concentration can decrease in case of liver damage. So, the use of food supplements with antioxidant capacity has been reported in the veterinary literature. In this case–control study, we tested a new supplement containing S-acetyl-glutathione (SAG), silybin, and other antioxidant ingredients in dogs affected by liver disease. After two weeks of supplement administration, we were able to report a significant increase in the level of erythrocyte GSH in the treated (TRT) group, nearly reaching the physiological limit at the end of the study. In addition, most of the key liver parameters are significantly reduced in the TRT group by the end of the trial. The results of this study support the effectiveness of the tested complementary feed, which may be helpful in managing dogs with liver conditions.

## 1. Introduction

The liver plays a role in regulating the endogenous antioxidant status, being the primary site for the synthesis of glutathione (GSH), the major intracellular antioxidant [1]. GSH is a thiol tripeptide formed by cysteine, glycine and glutamate. GSH is synthesized in all mammalian cells (primarily formed and stored in the liver) and it has different functions, such as the maintenance of the cellular redox state [2,3,4]. The important role of GSH as major antioxidant is to allow detoxification from the products of the aerobic metabolism (such as superoxide, hydrogen peroxide, and toxic oxygen radicals) and to protect tissues from cell damage [4,5]. GSH deficiency has been reported in people affected by many diseases, including liver disease, diabetes mellitus, renal failure, sepsis, and acute pancreatitis [3]. In veterinary medicine, dogs and cats can have decreased liver and blood GSH concentrations in cases of liver disease [1,5,6]. Consequently, therapeutic interventions aiming to overcome the GSH deficiency are recommended by clinicians. The use of several food supplements with antioxidant activities (such as S-adenosyl-methionine (SAME), *curcumin*, phosphatidylcholine, ursodeoxycholic acid, glycine, N-acetylcysteine (NAC), and silymarin) has been reported in the veterinary literature [7,8,9]. In particular, silymarin is a relevant natural ingredient with hepatoprotective function and antioxidant, immunomodulatory, anti-inflammatory, and choleretic proprieties in both human and veterinary medicine [7,10,11]. Silymarin is a flavonoid extracted from the milk thistle *Silybum marianum*, and it is composed of multiple flavonolignans, including the most active constituent—silybin [12].

In human medicine a valid approach to increase the endogenous GSH is the use of the S-acetyl-glutathione (SAG), which is a GSH precursor and originates from the fermentation of *Saccharomyces cerevisiae* [13]. SAG is also a potential hepatoprotective agent preventing oxidative liver damage [13]. However, no studies on dogs and cats regarding the use of SAG have been reported yet.

This case–control trial has been performed to test the hepatoprotective effect and the variation of erythrocyte GSH level when administering a new dietary supplement containing SAG and silybin associated with other well-known antioxidant components (orange bioflavonoid, vitamin B2, vitamin B12, vitamin E) in dogs with a diagnosis of liver disease.

## 2. Materials and Methods

This case–control study included a total of 24 adult dogs with a diagnosis of liver disease (cholangitis/cholangiohepatitis). The inclusion of the subjects was based on clinical and ultrasonographic signs, hepatic needle aspirate cytology, and hematobiochemical analysis. Dogs were excluded who presented other concomitant metabolic diseases or disorders potentially impacting the liver function (such as diabetes mellitus, gastritis, inflammatory bowel disease, chronic kidney disease, Cushing’s syndrome). Dogs administered with hepatoprotective products during the 30 days before the enrolment were also excluded.

All dogs were treated at the beginning of the study with antibiotics (oral amoxicillin-clavulanate BID 12.5–25 mg/kg), anti-inflammatory drugs (oral prednisolone SID 0.5 mg/Kg), and intravenous fluid therapy, as a supportive care for rehydration and the correction of electrolyte concentration if needed. All the animals enrolled had a diet based on Vetsolution monge epatic (50%) and Vetsolution digest (50%) from at least 14 days before starting the trial.

Twelve dogs were randomly assigned to the treatment group (TRT; male *n*= 5, female *n* = 7, mean age 6.8 years) and received the tested supplement (Table 1) at a dose of one tablet/15 kg BW, while the other 12 dogs were designated as the control group (CTR; male *n* = 6, female *n* = 6, mean age 6.7 years) and did not receive the supplemented diet.

For determining the concentration of the erythrocyte GSH, heparinized whole blood aliquots were frozen at −80 °C immediately after collection and then analyzed using a commercial assay kit (Ransel e Ransod, Randox Laboratories Ltd., Crumlin, UK) on an automated analyzer (RX Daytona™; Randox Laboratories Ltd., Crumlin, UK).

The effect of the supplement on the erythrocyte GSH and other blood parameters (total proteins (TP), albumins (ALB), alanine transaminases (ALT), alanine aminotransferases (AST), alkaline phosphatases (ALP), gamma–glutamyl transferase (GGT), bilirubin (BIL), and triglycerides (TRI)) was tested using a regression model built as a generalized linear mixed model (GLMM) with Gaussian likelihood (R software). The model included a non-linear variable describing the link between each time point within and between the CTR and TRT group, sex, body weight (BW), and age. The model account accounts for repeated measurements (random effect) and the heterogeneity of individuals.

## 3. Results and Discussion

The product under study was well tolerated by all the animals. During the trial no adverse effects (such as vomiting and diarrhea) were observed, confirming the safety of the product, and no dogs were excluded during the trial.

No significant alteration of blood counts (values within normal ranges) and no evidence of other diseases was reported based on the biochemical parameters evaluated at the baseline (Appendix A).

Results from the performed statistical analysis show that the supplement had no or limited effects on some of the biochemical values relevant to monitor liver disease (TP, ALB, GLU, TRI, and PCR) (Table 2).

Interestingly, most of the key liver parameters (ALT, AST, ALP, GGT, and BIL) are significantly reduced from T4 in the TRT group but not in the CTR group, as showed by the model results (Table 2). Indeed, the hepatoprotective activity of our tested supplement could be the result of the synergic effect of the included ingredients, the therapeutic proprieties of which had also pointed out in other research studies in animals. First of all, the efficacy of a supplement based on silybin with hepatoprotective proprieties was reported in a previous study on cats [12]. Then, a recently published review confirmed the reduction of ALT and GPT enzymes in dogs with hepatopathy following the administration of silybin [7]. In addition, the use of silybin for the treatment of hepatobiliary disease has also been reported in a study on cats, highlighting its antioxidant, anti-inflammatory, and anti-fibrotic capacities [12]. Marchegiani and colleagues [7] also reported a few case studies in which a combination of different active ingredients were selected and used (including silybin) to treat liver disease in dogs and cats.

Moreover, Di Paola and colleagues [13] described the effect of another of our ingredients, SAG, as its administration was functional to attenuating liver damage, to reduce liver fibrosis, and to improve hepatic function with a decrease in ALT and AST levels in humans. It has to be noted that the use of SAG derived from the fermentation of *Saccharomyces cerevisiae* in our supplement is a novelty in veterinary medicine as, to our knowledge, no data on its use in companion animals with liver conditions have yet been published. SAG not only has hepatoprotective proprieties, but it is also involved in GSH regulation. It is a GSH precursor and is more stable than GSH itself in plasma, being taken up directly by cells and later converted to GSH [13,14]. The oral administration of GSH itself is not significantly enhanced in plasma, while SAG is taken up by cells and later converted to GSH [13], and its absorption happens via the intestinal wall [13,15]. GSH is an important endogenous antioxidant that has been demonstrated to be significantly lower in the livers of dogs and cats with hepatic diseases [1,5]. This is in agreement with our findings as all our enrolled dogs showed low erythrocyte GSH levels (under the minimum physiological level, 300 Ug/Hb) at the beginning of the trial. In our study, results from the GLMM model showed a significant increase in erythrocyte GSH levels, even from T2 in the TRT group, and nearly reached the minimum physiological limit (300 Ug/Hb) at the end of the treatment. This could be the result of a joint effect of the previously mentioned SAG and of silybin, vitamin B2, vitamin B12, and vitamin E, which could all have potentially contributed to the increase in erythrocyte GSH [16,17,18,19]. In particular, silymarin, which has silybin as one of its flavonolignans, induced the hepatic synthesis of GSH by increasing cysteine availability in mice [16]. Moreover, vitamin B2 was demonstrated to play a role in the activities of GSH reductase and related antioxidant enzymes [17]. Regarding vitamin B12, it is a cofactor for the enzyme GSH reductase, a catalyst for converting the oxidized form of GSH back to its reduced active form [18]. Finally, Vitamin E was demonstrated to increase cellular GSH concentrations, as it displays radical-scavenging antioxidant activity [19].

This study has some limitations. First, the overall number of dogs included is relatively small and no placebo was given in the control group, thus affecting the quality of the study design.

## 4. Conclusions

To our knowledge, this is the first study on dogs with liver disease that tested a dietary supplement with a combination of natural antioxidants (SAG, silybin, orange bioflavonoid, vitamin B2, vitamin B12, and vitamin E) and was able to show hepatoprotective effects and a significant increase in erythrocyte GSH levels. The tested complementary feed may represent an effective aid in managing liver disease in dogs.

## Figures and Tables

**Table 1 vetsci-10-00131-t001:** Ingredients of the tested supplement.

Supplement		
Ingredients	g/100 g	mg/Tablet
Vitamin E	15.975	319.50
Orange bioflavonoid	8.0250	160.50
Vitamin B12	3.0000	60.00
Vitamin B2	2.0025	40.05
Silybin	2.0000	40.00
S-acetyl-L-glutathione	1.4250	28.50
Vitamin B1 hydrochloride	0.6375	12.75
Vitamin B6 hydrochloride	0.4350	8.70
Excipients	39.5000	790
Total	100.0000	2000.00

**Table 2 vetsci-10-00131-t002:** Mean and 95% Confidence Interval (95% CI) resulted from the biochemical analyses (total proteins (TP), albumins (ALB), glucose (GLU), alanine transaminases (ALT) alanine aminotransferases (AST), alkaline phosphatases (ALP), gamma-glutamyl transferase (GGT), bilirubin (BIL), triglycerides (TRI), c- reactive protein (PCR), and erythrocyte glutathione concentration (GSH)) in the control (CTR) and treated (TRT) group in the study period. Times: T0 (day 0), T1 (day 7), T2 (day 14), T3 (day 21), T4 (day 28), and T5 (day 35).

Time	TP (5.4–7.5 g/dL)		ALB (2.3–3.9 g/dL)		GLU (67–132 mg/dL)	
	CTR	TRT	CTR	TRT	CTR	TRT
T0	6.7 (6.6; 6.8)	6.6 (6.5; 6.7)	3.2 (3.2; 3.2)	3.2 (3.2; 3.3)	96 (95.1; 96.8)	96.2 (95.4; 97.1)
T1	6.6 (6.5; 6.7)	6.5 (6.4; 6.6)	3.2 (3.2; 3.2)	3.2 (3.2; 3.3)	95.4 (94.5; 96.3)	95.5 (94.7; 96.4)
T2	6.6 (6.5; 6.7)	6.6 (6.5; 6.7)	3.2 (3.2; 3.3)	3.2 (3.2; 3.3)	95.5 (94.7; 96.4)	95.2 (94.3; 96.1)
T3	6.6 (6.5; 6.7)	6.6 (6.5; 6.7)	3.2 (3.2; 3.3)	3.2 (3.2; 3.3)	94.8 (94; 95.7)	95.1 (94.2; 95.9)
T4	6.6 (6.5; 6.7)	6.7 (6.6; 6.8)	3.2 (3.2; 3.3)	3.2 (3.2; 3.3)	94.7 (93.9; 95.6)	94.5 (93.6; 95.3) *
T5	6.6 (6.5; 6.7)	6.7 (6.6; 6.8)	3.2 (3.2; 3.3)	3.2 (3.2; 3.3)	94.8 (94; 95.7)	94.9 (94.1; 95.8)
	ALT (0–40 UI/L)		AST (0–40 UI/L)		ALP (20–150 UI/L)	
Time	CTR	TRT	CTR	TRT	CTR	TRT
T0	303.4 (278.4; 328.4)	339 (314; 364.1)	285.8 (262.7; 308.9)	332.3 (309.1; 355.4)	219.1 (196.1; 242.1)	282.4 (259.4; 305.4)
T1	305 (279.9; 330)	335.7 (310.7; 360.8)	288 (264.9; 311.2)	328.8 (305.7; 351.9)	262.1 (239.2; 285.1)	280.5 (257.6; 303.5)
T2	302 (277; 327.1)	319.9 (294.9; 345)	307.1 (284; 330.2)	315 (291.9; 338.1) ^§^	256.7 (233.7; 279.6)	268.0 (245.0; 291.0) ^§^
T3	293.6 (268.5; 318.6)	292.8 (267.7; 317.8)	305.7 (282.6; 328.8)	290.9 (267.8; 314)	254.3 (231.3; 277.2)	246.4 (223.4; 269.3)
T4	292.3 (267.3; 317.3)	263.5 (238.4; 288.5) *	301.6 (278.5; 324.7)	263.3 (240.2; 286.4) *	252.4 (229.4; 275.3)	222.8 (199.9; 245.8) *
T5	297.8 (272.8; 322.8)	236.9 (211.8; 261.9) *	301.6 (278.4; 324.7)	238.4 (215.3; 261.5) *	247.9 (225; 270.9)	201.5 (178.5; 224.5) *
	GGT (2–8 UI/L)		BIL Tot (0–0.7 mg/dL)		TRI (23–110 mg/dl)	
Time	CTR	TRT	CTR	TRT	CTR	TRT
T0	23.8 (21.8; 25.9)	27.1 (25.1; 29.1)	2.1 (1.9; 2.3)	2.4 (2.2; 2.6)	138.3 (137.1; 139.6)	138.5 (137.2; 139.7)
T1	23.9 (21.8; 25.9)	26.8 (24.7; 28.8)	2.1 (1.9; 2.2)	2.3 (2.2; 2.5)	138.3 (137.1; 139.6)	138.3 (137; 139.6)
T2	23.9 (21.9; 25.9)	25.5 (23.4; 27.5)	2.1 (1.9; 2.3)	2.2 (2.1; 2.4)	138.1 (136.8; 139.4)	138.1 (136.9; 139.4)
T3	23.2 (21.2; 25.2)	23.3 (21.3; 25.3)	2.2 (2; 2.4)	2.1 (1.9; 2.2)	138.2 (136.9; 139.5)	138.1 (136.8; 139.3)
T4	22.7 (20.6; 24.7)	21 (18.9; 23) *	2.2 (2; 2.4)	1.9 (1.7; 2) *	138.2 (136.9; 139.5)	138.2 (136.9; 139.5)
T5	22.5 (20.5; 24.6)	18.8 (16.8; 20.9) *	2.2 (2.1; 2.4)	1.7 (1.5; 1.9) *	138.2 (136.9; 139.5)	138.2 (136.9; 139.4)
	PCR (0–1.5 mg/L)		GSH (300–700 U/g Hb)			
Time	CTR	TRT	CTR	TRT		
T0	1 (0.9; 1)	0.9 (0.9; 1)	266 (249.8; 282.2)	231.4 (215.2; 247.6)		
T1	0.9 (0.9; 1)	0.9 (0.9; 0.9)	268.5 (252.3; 284.7)	250.6 (234.4; 266.8)		
T2	0.9 (0.9; 1)	0.9 (0.9; 0.9)	264.1 (247.9; 280.3)	289.4 (273.2; 305.6) *^§^		
T3	0.9 (0.9; 0.9)	0.9 (0.9; 0.9)	267.6 (251.4; 283.8)	291.1 (274.9; 307.3) *		
T4	0.9 (0.8; 0.9) *	0.9 (0.9; 1)	272 (255.8; 288.2)	291.9 (275.7; 308.1) *		
T5	0.9 (0.8; 0.9)	0.9 (0.9; 0.9)	264.2 (248; 280.4)	293 (276.8; 309.2) *		

* Significant differences from T0 within group (*p* < 0.05), ^§^ Significant differences between the two groups at a specific time point (*p* < 0.05). Minimum and maximum values of the tested parameters are indicated in parentheses.

## Data Availability

The data that support the findings of this study are available on request from the corresponding author (F.P.).

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
