# Peer review of "Antioxidant Effect of a Dietary Supplement Containing Fermentative S-Acetyl-Glutathione and Silybin in Dogs with Liver Disease"

_vetsci, 2023, doi:10.3390/vetsci10020131_

Round 1
Reviewer 1 Report
The manuscript is well designed.
The author reported that dietary supplement with a combination of natural antioxidants (SAG, Sylibin, Orange bioflavonoid, vitamin B2, vitamin B12, and vitamin E) can show hepatoprotective effects and significant increase in the GSH levels. But a few points need to be clarified from the author.
1.Why are the numbers of males and females inconsistent in the treatment group?
2.How do you see the potential for this drug? There are many kinds of antioxidant substances, including SOD, Trx, etc., why the authors choose GSH?
3. The drug formulation in this study had many complex components, but the title only emphasized S-acetyl-glutathione and Sylibin, and even only GSH in the keywords. This part needs to be explained by the author, because the effect may be a combination, rather than a single drug.
4.The font in some tables needs to be consistent throughout the whole manuscript.
5. The Abstract should be rewritten. The application prospect of the drug should be mentioned in the abstract.
6. Keyword: Pet should be revised as dogs.
Author Response
Comments and Suggestions for Authors
The manuscript is well designed.
The author reported that dietary supplement with a combination of natural antioxidants (SAG, Sylibin, Orange bioflavonoid, vitamin B2, vitamin B12, and vitamin E) can show hepatoprotective effects and significant increase in the GSH levels. But a few points need to be clarified from the author.
1.Why are the numbers of males and females inconsistent in the treatment group?
AU: Thank you for the question. The dogs were randomly assigned to the control and treated group. Age, sex and weight were used in the statistical model to adjust the estimates.
2.How do you see the potential for this drug? There are many kinds of antioxidant substances, including SOD, Trx, etc., why the authors choose GSH?
AU: Thank you for the comment. We would like to specify that the product tested is not a drug but a complementary feed with natural ingredients. There is a range of ingredients listed in the register of additives we can combine in specific formulations. In this case, we wanted to test the effectiveness of multiple antioxidants when given in the same product focusing our attention to the specific effect on erythrocyte GSH and of course, on other relevant liver blood parameters.
- The drug formulation in this study had many complex components, but the title only emphasized S-acetyl-glutathione and Sylibin, and even only GSH in the keywords. This part needs to be explained by the author, because the effect may be a combination, rather than a single drug.
AU: Thank you for the note. We wanted to highlight the presence in the formulation of two of the ingredients as they are the novelty of this product with proven effects on GSH levels and key liver blood parameters. As we have also underlined in the text, the use of SAG derived from the fermentation of Saccharomyces cerevisiae, is a novelity in veterinary medicine as, at our knowledge, no data on its use in companion animals with liver conditions has been published yet. In addition, there is evidence of the effectiveness of Sylibin in pets with liver condition. We have also discussed that the results of the study could be determined by the joint effect of SAG, Sylibin, Vitamin B2, Vitamin B12, and Vitamin E which can all have potentially contributed to the increase of erytrocite GSH and improved other blood parameters.
4.The font in some tables needs to be consistent throughout the whole manuscript.
AU: Thank you. We corrected the tables.
- The Abstract should be rewritten. The application prospect of the drug should be mentioned in the abstract.
AU: The abstract was reviewed
- Keyword: Pet should be revised as dogs.
AU: Thank you for the suggestion- done.
Reviewer 2 Report
Dear authors, the study certainly has an interesting objective but requires in-depth analysis and revision.
There are several errors in the text, so it is recommended to re-read carefully.
The most common limits/questions are:
- inadequate sample
- the causes of hepatopathy in the selected patients have not been defined/described (and how they could influence the response to therapy)
- statistical analysis should be improved
- in addition to the compliance with national laws and the use of an informed consent, has the study also been approved by an ethics committee?
Author Response
Comments and Suggestions for Authors
Dear authors, the study certainly has an interesting objective but requires in-depth analysis and revision.
There are several errors in the text, so it is recommended to re-read carefully.
The most common limits/questions are:
- inadequate sample
AU: Thank you for the comment. We are aware of the limited sample size and we mentioned that in the limitation of the study. However, the results are very promising and further studies with a larger sample will be conducted to consolidate the results.
- the causes of hepatopathy in the selected patients have not been defined/described (and how they could influence the response to therapy)
AU: Thank you for the comment. Our included cases were diagnosed with cholangitis/ cholangiohepatitis based on ultrasonographic signs, hepatic needle aspirate cytology and hematobiochemical analysis. We decided not to separate in two the samples as the number of subjects were limited.In addition, we think that the results of the study would have not been significantly affected by this difference. We will take into account this observation when planning future trials on this supplement.
- statistical analysis should be improved
AU: Thank you for the comment. We realized the statistical analysis was not described in details in the methods. We added this part as we think that the statistical model used in this study is quite sophisticated and it allows to test the effect of the supplement on erythrocyte GSH and other blood parameters. The model was built as a generalized linear mixed model (GLMM) with Gaussian likelihood. The model included a non-linear variable describing the link between each time point within and between the CTR and TRT group, sex, BW and age. The identification of the subject was included in the model as a random effect to account for repeated measurements and the heterogeneity of individuals.
- in addition to the compliance with national laws and the use of an informed consent, has the study also been approved by an ethics committee?
AU: Thank you for the comment. We had the Approval from the Ethical Committee of the University but unfortunately we are still waiting for the final document to arrive by the end of January. We have already informed the Editorial Office of the Journal.
Reviewer 3 Report
This is a fascinating noble study with appropriate experiments. GSH is a powerful endogenous antioxidant, which positively affects many diseases in animals and humans. The study is well designed with proper interpretation of results. The limitation of the study is the small sample size mentioned in the article but still, this study provides a proof of concept for further study. Overall, this research finding is relevant to veterinary sciences readers. I am supportive of this paper however, some aspects below will need to be addressed before it is considered for publication.
The abstract can be little elaborated with important results findings.
Multiple spelling mistake/typos like dicrease, racommended, etc needs to be corrected.
The introduction looks good but missing important supplements (Glycine and n-acetylcysteine) that correct GSH deficiency in humans and mice (PMID: 35975308, PMID: 35268089)
The methods should be written in detail. Most readers of the journal will find it difficult to follow and cite these methods like GSH measurement methods not described.
The units of ALT, AST, etc. in the supplementary file are missing, and check the data with commas in place of decimals.
The results and findings look promising and well-discussed.
Author Response
Comments and Suggestions for Authors
This is a fascinating noble study with appropriate experiments. GSH is a powerful endogenous antioxidant, which positively affects many diseases in animals and humans. The study is well designed with proper interpretation of results. The limitation of the study is the small sample size mentioned in the article but still, this study provides a proof of concept for further study. Overall, this research finding is relevant to veterinary sciences readers. I am supportive of this paper however, some aspects below will need to be addressed before it is considered for publication.
The abstract can be little elaborated with important results findings.
AU: Thank you. We amended the abstract.
Multiple spelling mistake/typos like dicrease, racommended, etc needs to be corrected.
AU: Thank you. We checked the text.
The introduction looks good but missing important supplements (Glycine and n-acetylcysteine) that correct GSH deficiency in humans and mice (PMID: 35975308, PMID: 35268089)
AU: Thank you. We added the reference on mice in the introduction section.
The methods should be written in detail. Most readers of the journal will find it difficult to follow and cite these methods like GSH measurement methods not described.
AU: Thank you. We amended the methods.
The units of ALT, AST, etc. in the supplementary file are missing, and check the data with commas in place of decimals.
AU: Thank you. We amended S1 Table
The results and findings look promising and well-discussed.
AU: Thank you.
Round 2
Reviewer 2 Report
Dear author, thank you for your reply and for applying the corrections.
Line 92: please check the name of the products used for the diet